# K_Ca_1.1 K^+^ Channel Inhibition Overcomes Resistance to Antiandrogens and Doxorubicin in a Human Prostate Cancer LNCaP Spheroid Model

**DOI:** 10.3390/ijms222413553

**Published:** 2021-12-17

**Authors:** Susumu Ohya, Junko Kajikuri, Kyoko Endo, Hiroaki Kito, Miki Matsui

**Affiliations:** Department of Pharmacology, Graduate School of Medical Sciences, Nagoya City University, Nagoya 467-8601, Japan; kajikuri@med.nagoya-cu.ac.jp (J.K.); k.endo@med.nagoya-cu.ac.jp (K.E.); kito@med.nagoya-cu.ac.jp (H.K.); miki.matsui.4238@gmail.com (M.M.)

**Keywords:** cancer spheroid model, K_Ca_1.1, prostate cancer, cancer stemness, antiandrogen resistance, MDM2

## Abstract

Several types of K^+^ channels play crucial roles in tumorigenicity, stemness, invasiveness, and drug resistance in cancer. Spheroid formation of human prostate cancer (PC) LNCaP cells with ultra-low attachment surface cultureware induced the up-regulation of cancer stem cell markers, such as NANOG, and decreased the protein degradation of the Ca^2+^-activated K^+^ channel K_Ca_1.1 by down-regulating the E3 ubiquitin ligase, FBXW7, compared with LNCaP monolayers. Accordingly, K_Ca_1.1 activator-induced hyperpolarizing responses were larger in isolated cells from LNCaP spheroids. The pharmacological inhibition of K_Ca_1.1 overcame the resistance of LNCaP spheroids to antiandrogens and doxorubicin (DOX). The protein expression of androgen receptors (AR) was significantly decreased by LNCaP spheroid formation and reversed by K_Ca_1.1 inhibition. The pharmacological and genetic inhibition of MDM2, which may be related to AR protein degradation in PC stem cells, revealed that MDM2 was responsible for the acquisition of antiandrogen resistance in LNCaP spheroids, which was overcome by K_Ca_1.1 inhibition. Furthermore, a member of the multidrug resistance-associated protein subfamily of ABC transporters, MRP5 was responsible for the acquisition of DOX resistance in LNCaP spheroids, which was also overcome by K_Ca_1.1 inhibition. Collectively, the present results suggest the potential of K_Ca_1.1 in LNCaP spheroids, which mimic PC stem cells, as a therapeutic target for overcoming antiandrogen- and DOX-resistance in PC cells.

## 1. Introduction

Androgen deprivation therapy (ADT) is the standard care for the initial management of advanced and metastatic prostate cancer (PC); however, progression to castration-resistant PC (CRPC) occurs within a few years of the initiation of ADT [1]. The agents currently available for the treatment of CRPC include (1) chemotherapy agents, such as docetaxel (DTX), and (2) antiandrogens, including bicalutamide (BCT) and enzalutamide (EZT) [2,3]. However, PC patients with metastatic CRPC eventually become resistant to these agents [2,3].CRPC under ADT causes genetic changes in androgen receptors (AR), such as overexpression, mutations, and splice variants [4]. Moreover, CRPC is also driven from PC stem cells (PCSCs), and the sustained prevention of AR protein expression by their degradation is one potential mechanism by which PCSCs acquire antiandrogen resistance [5]. Recently, poly (ADP-ribose) polymerase (PARP) inhibitors, such as olaparib, novel AR antagonists, such as darolutamide, and radio-ligand therapy with lutetium prostate-specific membrane antigen (Lu-PSMA) have emerged as potentially effective therapeutic options in patients with CRPC [6].

Multicellular tumor three-dimensional (3D) spheroid models are spherical self-assembled aggregates of cancer cells. They are a valuable tool for studying the tumor microenvironment (TME) in solid tumors and are also important for investigating the characteristics of CSCs, such as tumor initiation, metastasis, and resistance to chemotherapies (chemoresistance) [7,8]. Previous studies demonstrated that the expression of CSC markers, such as CD44 and NANOG, was upregulated in aggregated PC spheroid models and contributed to their acquisition of chemoresistance and antiandrogens [9,10]. Ultra-low attachment plates and dishes pre-coated with an ultra-hydrophilic polymer promoted the spontaneous formation of cancer spheroids.

Ion channels contribute to tumorigenicity, stemness, invasiveness, and chemoresistance in several cancers, including PC [11,12]. Recent studies have focused on the potential contribution of K^+^ channels to cancer progression in TME and to the acquisition of therapy resistance [13,14]. The large-conductance Ca^2+^-activated K^+^ channel K_Ca_1.1 encoded by KCNMA1 overexpresses in PC and enhances the proliferative and metastatic abilities of PC cells through modifications of Ca^2+^ signaling [15,16]. The expression of K_Ca_1.1 is regulated by transcriptional, epigenetic, and post-translational modifications in cancers [17,18]. The functional diversity of K_Ca_1.1 also arises from alternative splicing and modifications by non-pore-forming regulatory molecules, such as β and γ subunits [19].

Post-translational modifications by E3 ubiquitin ligases (E3s) play a crucial role in protein stabilization, and the dysregulation of ubiquitination is associated with cancer progression, metastasis, and therapy resistance [20]. In sarcoma spheroid models, the downregulation of F-box/WD repeat-containing protein 7 (FBXW7) decreased the protein degradation of K_Ca_1.1, which increased K_Ca_1.1 activity [21]. Cereblon (CRBN) plays an important role in the assembly and plasma membrane expression of functional K_Ca_1.1 [22] and also promotes AR protein degradation in androgen-sensitive human PC cell line, LNCaP [23]. In PCSCs, murine double minute 2 (MDM2) causes the sustained loss of AR protein expression by promoting the constant degradation of AR [24].

ATP-binding cassette (ABC) transporters, such as multidrug resistance (MDR/ABCB) and multidrug resistance-associated proteins (MRP/ABCC), are essential for the acquisition of chemoresistance [25]. MDR1 and MDR3 were previously shown to be involved in the acquisition of resistance to paclitaxel (PTX) and doxorubicin (DOX), respectively, while MRP1, MRP2, and MRP5 play a role in the acquisition of resistance to DOX and/or cisplatin (CIS) [26]. MRP4 is involved in the acquisition of resistance to DTX, but not to PTX, DOX, or CIS [26].

In the present study, human PC spheroid models with antiandrogen resistance were obtained from a LNCaP cell line *in vitro* using ultra-low attachment cultureware. The main objective of the present study is to elucidate the mechanisms underlying the acquired resistance to antiandrogens as it is overcome by the inhibition of K_Ca_1.1 using a LNCaP spheroid model.

## 2. Results

### 2.1. Increased K_Ca_1.1 Activity in the LNCaP Spheroid Model

The androgen-sensitive LNCaP functionally expresses K_Ca_1.1 and the auxiliary γ subunit, leucine-rich repeat-containing protein 26 (LRRC26) promotes the activation of K_Ca_1.1 at physiological voltages and Ca^2+^ levels [27]. Furthermore, the other commercially-available human PC cell lines, VCaP, PC-3, and DU-145, rarely expressed K_Ca_1.1. Spheroidal aggregates of LNCaP cells were formed 5–7 days after cell seeding onto the ultra-low attachment cultureware (day 7, lower panel, Figure 1A). Hyperpolarizing responses induced by NS1619 (1 μM), a K_Ca_1.1 activator, were significantly larger in isolated cells from 3D spheroids on day 7 (‘3D’) than in those from 2D adherent cell monolayers (‘2D’) (*p* < 0.05) (Figure 1B,C). A pretreatment with the selective K_Ca_1.1 blocker, paxilline (PAX) (1 μM), almost reduced NS1619-induced hyperpolarizing responses (Appendix A). The expression level of the K_Ca_1.1 protein increased by approximately four-fold in lipid-raft fractions (RIPA-insoluble, ultra-RIPA-soluble fractions) of LNCaP spheroids (*p* < 0.05) (Figure 1D,E), without changes in that of the K_Ca_1.1 transcript (*p* > 0.05) (Figure 1F). Similarly, the flow cytometric analysis using the Alexa Fluor 488-conjugated anti-K_Ca_1.1 antibody showed that the relative mean intensity of Alexa Fluor 488 was approximately 4.5-fold higher in LNCaP cells from ‘3D’ than ‘2D’ (Appendix A). Moreover, the expression levels of the cancer stemness markers, NANOG [28], CD44, and KLF4, were markedly higher in 3D spheroids than in 2D monolayers (Figure 2). Among six auxiliary β and γ subunits, LRRC26 was predominantly expressed in both LNCaP monolayers and spheroids (Appendix A). We measured outward K^+^ currents elicited in 3D-cultured LNCaP cells by depolarizing voltage steps between −80 and +60 mV from a holding potential of −60 mV using whole-cell patch-clamp recordings. Outward K^+^ currents were completely blocked by the application of 1 μM PAX (Appendix A).

### 2.2. Acquired Resistance of LNCaP Spheroids to Antiandrogens and DOX Overcome by K_Ca_1.1 Inhibition

The optimization of the initial cell seeding density is essential for the WST-1 cell proliferation assay. Especially, it affects various parameters, such as metabolic activity and drug penetration in a 3D spheroid. Under the condition described in Section 4.2. (10^5^ cells/mL at day 0), LNCaP cells cultured in a 2D monolayer almost doubled every 24 h from day 0 to day 3. Moreover, the metabolic activity of the WST-1 reagent in cells cultured in a 3D spheroid was reduced by almost 30% in 2 days; however, it was kept for the additional 7 days. We further examined the WST-1 assay in 3D spheroids with the different initial cell densities (2 × 10^4^, 5 × 10^4^, 10^5^, 2 × 10^5^, and 5 × 10^5^ cells/mL); however, the drug sensitivities were almost same among them. As shown in Figure 3A–D, LNCaP spheroids acquired chemoresistance to DTX, PTX, DOX, and CIS following exposure for 48 h (*p* < 0.01). In LNCaP spheroids, the resistance to DOX alone was reversed by the pretreatment with 10 μM PAX for 24 h (*p* < 0.01) (Figure 3E–H). No significant changes in cell viability were found by the treatment with PAX alone for 48 h (1.032 ± 0.029 in relative cell viability (*n* = 5, *p* > 0.05)).

LNCaP spheroids also exhibited resistance to the antiandrogens BCT and EZT (*p* < 0.01) (Figure 4A,B). The PAX pretreatment significantly reversed the antiandrogen resistance acquired by LNCaP spheroids (*p* < 0.01) (Figure 4C,D). There results suggest that the inhibition of K_Ca_1.1 may represent a novel strategy for overcoming antiandrogen resistance in patients with CRPC.

Among the 10 ABC transporter candidates (MDR1, MDR3, MRP1-6, and ABCG1-2) possibly involved in chemoresistance, the MRP1, 3, 4, and 5 transcripts were expressed at high levels in LNCaP spheroids, with the levels of transcripts other than MRP4 significantly increasing with the spheroid formation (Figure 5A–D). The expression levels of the other candidates were much lower in LNCaP spheroids (less than 0.002 in arbitrary units to ACTB). As shown in Figure 5E–H, the PAX treatment for 24 h significantly reversed the expression of MRP5 only. Western blot (WB) examinations also showed that increases in MRP5 protein expression levels in LNCaP spheroids were significantly reduced by the PAX treatment (*p* < 0.05) (Appendix A). No significant changes in MRP1, MRP3, and MRP4 protein expression levels were found by the PAX treatment (Appendix A).

Furthermore, a co-treatment with the MRP4/5 inhibitor, MK571 and DOX overcame DOX resistance in LNCaP spheroids, whereas that with the selective MRP4 inhibitor, ceefourin 1 and DOX, did not (Figure 6C,G). Neither inhibitor overcame resistance to the other chemotherapy drugs (Figure 6). No significant changes in cell viability were observed with single treatments with MK571 or ceefourin 1 for 48 h (0.973 ± 0.031 and 0.965 ± 0.045, respectively, in relative cell viability (*n* = 5, *p* > 0.05)). These results strongly suggest that MRP5 plays a critical role in the acquisition and K_Ca_1.1 inhibition-induced overcoming of resistance by LNCaP spheroids to DOX.

### 2.3. Decreases in AR Protein Expression in Antiandrogen-Resistant-LNCaP Spheroids

We compared the gene and protein expression levels of AR between 2D monolayers and 3D spheroids of LNCaP cells using real-time PCR and WB examinations. The protein expression level of AR with a molecular weight of approximately 110 kDa decreased in lipid-raft fractions of LNCaP spheroids (*p* < 0.01) (Figure 7B,C), without changes in that of AR transcripts (*p* > 0.05) (Figure 7A). Consistent with the acquisition of resistance to antiandrogens being overcome by the PAX treatment (Figure 4), a pretreatment with 10 μM PAX significantly increased AR protein levels in LNCaP spheroids (*p* < 0.01) (Figure 7E,F), without changes in AR transcript levels (*p* > 0.05) (Figure 7D). The MRP4/5 inhibitor, MK571, did not overcome resistance to antiandrogens by LNCaP spheroids (Appendix A). These results suggest that AR protein degradation plays a role in the acquisition of antiandrogen resistance in LNCaP spheroids, and that K_Ca_1.1 inhibitors may prevent the activity and/or expression of E3s involved in AR degradation-mediated antiandrogen resistance by PCSCs.

### 2.4. Decrease in the Expression of the AR-Targeted Ubiquitin E3 Ligase, MDM2, in LNCaP Spheroids

Several E3s that degrade the AR protein have been identified in PC cells: SKP2 (S-phase kinase-associated protein 2), RNF6 (RING (really interesting new gene) finger protein 6), UBE3A (ubiquitin protein ligase E3A), SIAH2 (seven in absentia (Drosophila) homolog 2), CRBN, MDM2, NRIP (nuclear receptor interaction protein), and USP14 (ubiquitin-specific protease 14) [22,23,24,29]. The transcriptional expression levels of six candidates, except for MDM2 and CRBN, was not downregulated by the PAX treatment for 24 h (Appendix A). Moreover, the expression levels of the MDM2 and CRBN transcripts and proteins were inversely correlated with AR protein expression (2D vs. 3D, Figure 8A–E; vehicle-treated vs. PAX-treated, Figure 8F–J).

To elucidate the roles of CRBN and MDM2 in antiandrogen resistance through the promotion of AR protein degradation in LNCaP spheroids, we examined the effects of the pharmacological and siRNA-mediated inhibition of CRBN and MDM2 on acquired antiandrogen resistance. A pretreatment with the MDM2 inhibitor, nutlin-3a, for 24 h significantly reversed the resistance to BCT and EZT in LNCaP spheroids (*p* < 0.01) (Figure 9A), whereas that with the CRBN inhibitor, iberdomide, did not (*p* > 0.05) (Figure 9B). Similarly, the siRNA-mediated inhibition of MDM2 reversed the resistance to antiandrogens in LNCaP spheroids (*p* < 0.01) (Figure 9C), whereas that of siRNA-mediated CRBN inhibition did not (*p* > 0.05) (Figure 9D). Correspondingly, the siRNA-mediated inhibition of MDM2, but not CRBN, significantly increased AR protein levels in LNCaP spheroids (*p* < 0.05 in si-MDM2; *p* > 0.05 in si-CRBN) (Figure 9E–H). These results strongly suggest that MDM2 is responsible for the antiandrogen resistance induced by spheroid formation in LNCaP cells through the promotion of AR protein degradation, and also that the transcriptional repression of MDM2 mediated by the inhibition of K_Ca_1.1 overcame the antiandrogen resistance acquired by LNCaP spheroids through the increase in AR protein levels.

### 2.5. Involvement of the Ubiquitin E3 Ligase, FBXW7, in Enhancements in K_Ca_1.1 Activity Induced by LNCaP Spheroid Formation

FBXW7 promoted the protein degradation of K_Ca_1.1 in human breast cancer and sarcoma cell lines [18,21]. The protein expression levels of FBXW7 were markedly decreased by LNCaP spheroid formation (*p* < 0.01) (Figure 10A,B), and the siRNA-mediated inhibition of FBXW7 significantly increased the protein expression levels of K_Ca_1.1 in 2D-cultured LNCaP cells (*p* < 0.05) (Figure 10C,D). These results suggest that FBXW7 plays a role in K_Ca_1.1 protein degradation in LNCaP cells.

## 3. Discussion

Recent studies demonstrated that K^+^ channels are important contributors to drug resistance being overcome in solid cancers [14,30]. K_Ca_1.1 gene amplification plays an important role in PC progression [15]. The main results of the present study are as follows: (1) the inhibition of K_Ca_1.1 overcame acquired resistance to antiandrogens in LNCaP spheroids (Figure 4); (2) AR protein degradation by the upregulation of MDM2 was associated with the acquisition of antiandrogen resistance in LNCaP spheroids, and the pharmacological inhibition of K_Ca_1.1 reversed MDM2-mediated AR degradation, which overcame antiandrogen resistance (Figure 7, Figure 8 and Figure 9); and (3) the inhibition of K_Ca_1.1 overcame acquired resistance to DOX in LNCaP spheroids (Figure 3), and MRP5 was associated with both the acquisition of DOX resistance and it being overcome by the inhibition of K_Ca_1.1 (Figure 5 and Figure 6). We also found that K_Ca_1.1 proteins were overexpressed in LNCaP spheroids, together with a decrease in FBXW7-mediated K_Ca_1.1 protein degradation (Figure 1 and Figure 10). Collectively, the present results demonstrated the impact of K_Ca_1.1 on the resistance to antiandrogens and DOX in LNCaP spheroid models.

The E3, MDM2, promotes cancer stemness with an AR-negative signature in PCSCs by selectively degrading AR proteins [24]. In contrast, the loss of MDM2 promotes CSC differentiation by reversing the processes of PC stemness [24]. In the present study, several PCSC markers were overexpressed by LNCaP spheroid formation (Figure 2). NANOG has been implicated in the acquisition of cancer therapy resistance [31], and its overexpression in PC cells has been shown to reduce AR levels [32]. MDM2 and CRBN were included among the 6490 target genes of the NANOG transcription factor obtained from the chromatin immunoprecipitation (ChIP) enrichment analysis (ChEP) transcriptional factor targets dataset [33]. These findings suggest that NANOG is a transcription factor of MDM2 in PCSCs; however, further studies are needed to obtain direct evidence to show that MDM2 is a NANOG downstream gene in PCSCs. Additionally, several studies have shown that signal transducer and activator of transcription (STAT) signaling pathways play critical roles in the AR protein degradation and stabilization in PC cells [34,35]. Further studies will be needed to elucidate the possible involvement of K_Ca_1.1 in the regulation of STAT pathway-mediated AR protein degradation.

The present study indicated that acquired resistance to DOX in LNCaP spheroids was due to upregulated MRP5 expression (Figure 3C and Figure 5D). Consistent with these results, DOX resistance was reversed by a treatment with the MRP4/5 inhibitor without affecting the selective MRP4 inhibitor (Figure 6C,G). These results are in accordance with recent findings showing that MRP5 was associated with drug resistance to DOX, but not PTX or CIS [26,36]. Different from our recent study on sarcoma spheroids [21], the inhibition of K_Ca_1.1 did not affect the expression levels of MRP1 (Figure 5E) and Nrf2 (Appendix A) in LNCaP spheroids. In fibrosarcoma spheroids, the NANOG inhibition reverses DOX resistance [31]. As shown in Appendix A, the inhibition of K_Ca_1.1 downregulated the expression of the CSC markers in LNCaP spheroids. It currently remains unclear whether MRP5 is a NANOG downstream gene in PCSCs; however, it was not included among the 6490 target genes of the NANOG transcription factor obtained from the ChEP transcriptional factor targets dataset. Furthermore, Ji et al. (2021) recently showed that the high expression of MRP5 reduced the sensitivity of LNCaP spheroids to EZT [37]. However, the present study showed no significant changes in antiandrogen resistance in LNCaP spheroids treated with MK571 (Appendix A).

Consistent with human sarcoma spheroid models [21], K_Ca_1.1 protein degradation was suppressed by the downregulation of FBXW7 in LNCaP spheroids (Figure 10) [21]. Recent studies have focused on the role of mitochondrial ion channels in chemoresistance in cancers [38,39]. Mitochondrial K_Ca_1.1 may be responsible for the acquisition of resistance to antiandrogens and chemotherapies by PCSCs. A more detailed understanding of K_Ca_1.1-regulated resistance to antiandrogens and chemotherapies is needed for the development of novel treatment strategies for CRPCs associated with PCSCs.

The pro-inflammatory cytokines, IL-1β and IL-8, have been shown to repress AR mRNA expression in PC cells [40,41], and play a role in antiandrogen resistance in PC [42]. However, their levels in LNCaP spheroids were undetectable by an ELISA assay (less than 1 pg/mg protein). As shown in Figure 7D, the inhibition of K_Ca_1.1 in LNCaP spheroids did not affect the expression level of AR; however, IL-1β and/or IL-8-producing, tumor-infiltrating non-cancerous cells, such as tumor-associated macrophages cells in the TME, may be responsible for the AR transcriptional repression-mediated antiandrogen resistance in PCSCs.

In the TME of PC, tumor-infiltrating, non-cancerous cells, such as regulatory T cells, tumor-associated macrophages (TAMs), myeloid-derived suppressor cells (MDSCs), and cancer-associated fibroblasts (CAFs), play a critical role in immunosuppression [43]. A recent study showed that AR-expressing CAFs can affect PC progression and metastasis [44]. In addition, TAM-derived IL-6 and IL-8 are important for the regulation of AR expression and antiandrogen resistance [44]. Studies to shed light on the cellular heterogeneity of TME using accurate TME-mimicking cancer organoid models and patient-derived organoids will be needed to elucidate more mechanisms underlying AR protein degradation and antiandrogen resistance in PCSCs.

## 4. Materials and Methods

### 4.1. Materials and Reagents

The following chemicals and reagents were used: EZT, iberdomide, MK571 (MedChemExpress, Monmouth Junction, NJ, USA), BCT (Adooq Bioscience, Irvine, CA, USA), PAX, nutlin-3a (Cayman Chemical, Ann Arbor, MI, USA), NS1619, ceefourin 1 (Abcam, Cambridge, UK), docetaxel (DTX) (TCI, Tokyo, Japan), PTX, DOX hydrochloride, CIS, RPMI 1640 medium (FUJIFILM, Osaka, Japan), DiBAC_4_(3), WST-1 (Dojindo, Kumamoto, Japan), Select Pre-designed/Validated siRNAs as a negative control (Pre-designed, No. 1), CRBN (Pre-designed, ID#: s534902), FBXW7 (Pre-designed, ID#: s224356), MDM2 (Validated, ID#: s8630) and Lipofectamine^®^ RNAiMAX (Thermo Fisher Scientific, Waltham, MA, USA), and ReverTra Ace (ToYoBo, Osaka, Japan). The other chemicals used in the present study were from Sigma-Aldrich (St. Louis, MO, USA), FUJIFILM, and Nacalai Tesque (Kyoto, Japan), unless otherwise stated.

### 4.2. Cell Culture

The human PC LNCaP cell line was purchased from the RIKEN Cell Bank (Osaka, Japan). They were cultured in RPMI 1640 medium supplemented with 10 nM dihydrotestosterone, 10% fetal bovine serum, and a penicillin–streptomycin mixture. The cells were cultured at 37 °C in a humidified atmosphere containing 5% CO_2_. Flat-bottomed dishes and plates (Corning, Corning, NY, USA) were used for the 2D cell culture [21]. The PrimeSurface^®^ system (Sumitomo Bakelite, Tokyo, Japan) was used for the 3D spheroid culture. LNCaP cell suspensions were seeded onto a PrimeSurface 96U plate at 10^5^ cells/well, and then cultured for 7 days.

### 4.3. Real-Rime PCR

Total RNA was isolated from cancer cells using the conventional acid guanidinium thiocyanate-phenol-chloroform extraction method. The concentration and quality of RNA were confirmed using the microvolume spectrophotometer, NanoDrop One (Thermo Fisher Scientific). Reverse transcription was performed using ReverTra Ace with random hexanucleotides. Quantitative, real-time PCR was conducted using the Luna Universal qPCR Master Mix (New England Biolabs Japan, Tokyo, Japan) and the Applied Biosystems 7500 Fast Real-Time PCR System (Thermo Fisher Scientific). PCR primers of human origin were listed in Appendix A. Relative expression levels were calculated using the 2^−ΔΔCt^ method [21] and normalized to ACTB.

### 4.4. Measurements of K_Ca_1.1 Activity by Voltage-Sensitive Dye Imaging and Whole-Cell Patch Clamp Recordings

Membrane potential was measured using the fluorescent voltage-sensitive dye DiBAC_4_(3) [18,21]. Briefly, prior to fluorescence measurements, the cells were incubated in normal 2-[4-(2-Hydroxyethyl)-1-piperazinyl]ethanesulfonic acid (HEPES) buffer containing 100 nM DiBAC_4_(3) at room temperature for 20 min, and the cells were then continuously incubated in 100 nM DiBAC_4_(3) throughout the experiments. Hyperpolarizing responses induced by the K_Ca_1.1 activator, NS1619 (1 μM) (decrease in fluorescence intensity), were measured using an ORCA-Flash2.8 digital camera (Hamamatsu Photonics, Hamamatsu, Japan). Data collection and analyses were performed using the HCImage system (Hamamatsu Photonics). Images were measured every 5 s.

A whole-cell patch clamp was applied to 3D-cultured LNCaP cells using the Axon Patch Clamp System (Axopatch 200B amplifier, Molecular Devices, San Jose, CA, USA) at room temperature (23 ± 1 °C). Data acquisition and analyses of whole cell currents were performed using Axon Digidata 1550B plus HumSilencer and Axon pCLAMP Software Suite (Molecular Devices). Whole-cell currents were measured in the voltage-clamp mode and induced by 500-ms voltage steps, every 15 s, from −80 to +60 mV at a holding potential of −60 mV. The external solution was (in mM): 137 NaCl, 5.9 KCl, 2.2 KCl, 1.2 MgCl_2_, 14 glucose, and 10 HEPES, pH7.4. The pipette solution was (in mM): 140 KCl, 4 MgCl_2_, 3.2 CaCl_2_, 5 EGTA, 10 HEPES, and 2 Na_2_ATP, pH 7.2, with an estimated free Ca^2+^ concentration of 100 nM (pCa 7.0).

### 4.5. Western Blotting

Whole cell lysates were extracted using a radioimmunoprecipitation (RIPA) buffer. Lipid-raft-enriched protein fraction lysates were extracted using the ULTRARIPA kit for Lipid Rafts (BioDynamics Laboratory, Tokyo, Japan), according to the manufacturer’s instructions. Equal amounts of protein were subjected to SDS-PAGE and immunoblotting with anti-K_Ca_1.1 polyclonal (rabbit) (approximately 100 kDa) (1:750, APC-021, Alomone Labs, Jerusalem, Israel), anti-AR polyclonal (rabbit) (approx. 110 kDa) (1:750, C-19, Santa Cruz Biotechnology, SCB, Santa Cruz, CA, USA), anti-FBXW7 polyclonal (rabbit) (approx. 70 kDa) (1:2000, ABclonal, Tokyo, Japan), anti-CRBN polyclonal (rabbit) (approx. 55 kDa) (1:1500, ABclonal), anti-MDM2 polyclonal (rabbit) (approx. 95 kDa) (1:2000, ABclonal), anti-MRP1 polyclonal (rabbit) (approx. 250 kDa) (1:1500, Bioss Antibodies, Woburn, MA, USA), anti-MRP3 polyclonal (rabbit) (approx. 170 kDa) (1:1000, ABclonal), anti-MRP4 polyclonal (rabbit) (160–200 kDa) (1:2000, ABclonal), anti-MRP5 polyclonal (rabbit) (approx. 160 kDa) (1:1500, ABclonal), and anti-ACTB monoclonal (mouse) (43 kDa) (1:15,000, 6D1, Medical & Biological Laboratories, Nagoya, Japan) antibodies, and then incubated with anti-rabbit and mouse horseradish peroxidase-conjugated IgG (Merck Millipore, Darmstadt, Germany). Strong band signal for flotillin-1 (45 kDa, anti-flotillin polyclonal (rabbit) antibody, GeneTex, Alton Pkwy Irvine, CA, USA) protein, a lipid-raft marker, was detected in ultra-RIPA soluble fractions, but not RIPA soluble ones. Image detections and the quantitative analyses of the optical densities of protein band signals were performed as previously reported [21].

### 4.6. Immunocytochemistry

Fixed and non-permeabilized cells were stained with an anti-K_Ca_1.1 (extracellular) polyclonal antibody (rabbit) (APC-151, Alomone Labs) followed by an Alexa Fluor 488-conjugated secondary antibody (Thermo Fisher Scientific), and then analyzed by flow cytometry (FACSCanto II, BD Biosciences, San Jose, CA, USA). K_Ca_1.1 expression was expressed as means fluorescence intensity after the subtraction of that in cells stained by the secondary antibody [21].

### 4.7. Cell Viability Assay

Cell viability was assessed using the WST-1 assay [21]. Briefly, using a density of 10^5^ cells/mL, cells were cultured in duplicate in 96-well plates for 7 days (for the 3D culture) and 1 day (for the 2D culture). The cells were then treated with drugs used in chemotherapy (DTX, PTX, DOX, and CIS) and antiandrogens (BCT and EZT) for 48 h. Two hours after the addition of WST-1 reagent to each well, the absorbance was measured using the microplate reader SpectraMax 384 (Molecular Devices Japan, Tokyo, Japan) at a test wavelength of 450 nm and reference wavelength of 650 nm.

### 4.8. Targeted Gene Suppression by siRNA Transfection

Lipofectamine^®^ RNAiMAX reagent (Thermo Fisher Scientific) was used in the siRNA-mediated blockade of FBXW7, CRBN, and MDM2 according to the manufacturer’s protocol. Silencer Select Pre-designed/Validated siRNAs for the negative control (No. 1) (si-Cont), FBXW7 (si-FBXW7), CRBN (si-CRBN), and MDM2 (si-MDM2) were transfected into adherent monolayer LNCaP cells. Twenty-four hours later, si-CRBN- and si-MDM2-transfected cells were seeded onto PrimeSurface 96U. The expression levels of the target transcripts were assessed 3 days (for si-FBXW7 transfectant) and 7 days (for si-CRBN and si-MDM2 transfectants) after transfection using real-time PCR assay, resulting in more than 70% reduction.

### 4.9. Statistical Analysis

Statistical evaluation was performed with Statistical software XLSTAT (version 2013.1). Unpaired/paired Student’s t-tests with Welch’s correction or Tukey’s tests were used to assess the significance of differences between two groups and among multiple groups. Results with a *p* value of less than 0.05 were considered to be significant. Data were presented as means ± SEM.

## 5. Conclusions

In conclusion, the present results suggest that K_Ca_1.1 may be a key modulator of antiandrogen resistance in androgen-sensitive, K_Ca_1.1 gene-amplified PC, and that K_Ca_1.1 inhibitors may overcome acquired antiandrogen resistance in PCSCs. Furthermore, K_Ca_1.1 played a vital role in the overcome of DOX resistance by downregulating MRP5 in LNCaP spheroids, suggesting the potential of K_Ca_1.1 inhibitors as an effective therapeutic intervention in combination with antiandrogens. The targeting of K_Ca_1.1 is promising for suppressing the progression of CRPC associated with PCSCs through the regulation of MDM2-mediated AR degradation.

## Figures and Tables

**Figure 1 ijms-22-13553-f001:**
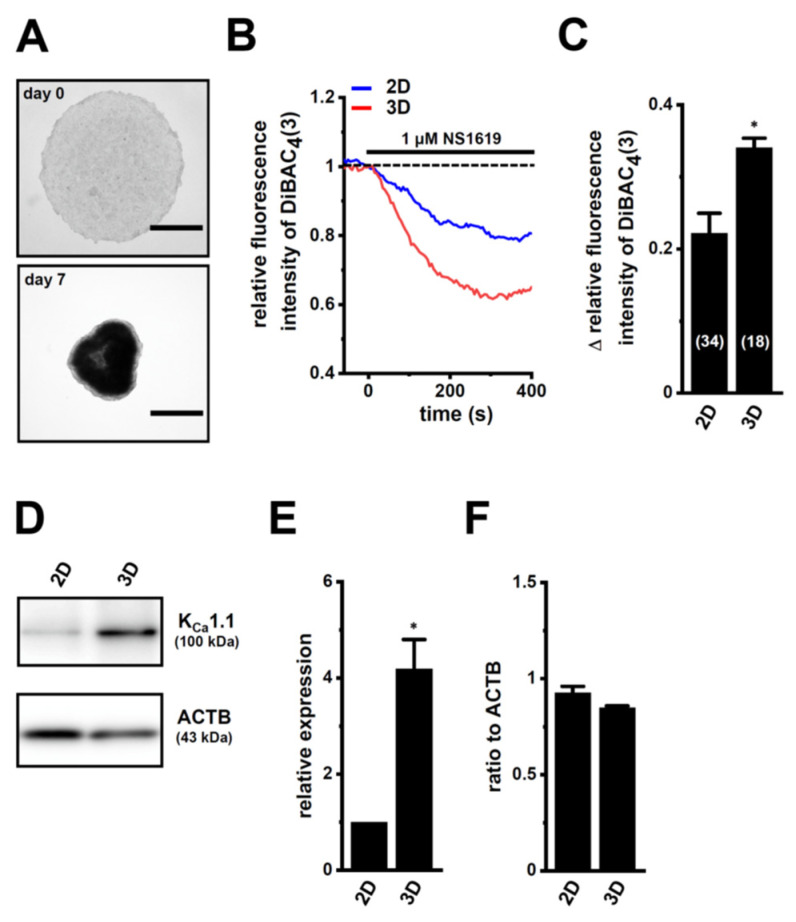
Comparison of K_Ca_1.1 expression and activity between LNCaP cells cultured as 2D monolayers and 3D spheroids. (**A**): Phenotypic properties of LNCaP cells cultured in ultra-low attachment PrimeSurface 96U plates (upper panel: on day 0; lower panel: on day 7). Brightfield images were obtained with the Axio Observer Z1 microscope system (Carl Zeiss, Oberkochen, Germany). Bars show 500 μm. (**B**): Time course of the voltage-sensitive fluorescent dye imaging of K_Ca_1.1 activator (1 μM NS1619)-induced hyperpolarizing responses in isolated cells from ‘2D’ monolayers and ‘3D’ spheroids of LNCaP. The fluorescent intensity of DiBAC_4_(3) before the application of NS1619 is expressed as 1.0. Images were measured every 5 s. (**C**): Summarized results of NS1619-induced hyperpolarizing responses in cells isolated from at least three different batches in each group. Cell numbers used in experiments are shown in parentheses. The values for fluorescent intensity were obtained by measuring the average for 1 min (12 images). (**D**): K_Ca_1.1 protein expression in the lipid-raft-enriched protein lysates of both groups. Blots were probed with anti-K_Ca_1.1 (approximately 100 kDa, upper panel) and anti-ACTB (43 kDa, lower panel) antibodies. (**E**): Summarized results were obtained as the optical density of K_Ca_1.1 and ACTB band signals. After compensation for the optical density of the K_Ca_1.1 protein band signal with that of the ACTB signal, the K_Ca_1.1 signal in ‘2D’ was expressed as 1.0 (*n* = 4 for each). (**F**): Real-time PCR examination of K_Ca_1.1 in both groups (*n* = 4 for each). Expression levels were shown as a ratio to ACTB. *: *p* < 0.05 vs. ‘2D’.

**Figure 2 ijms-22-13553-f002:**
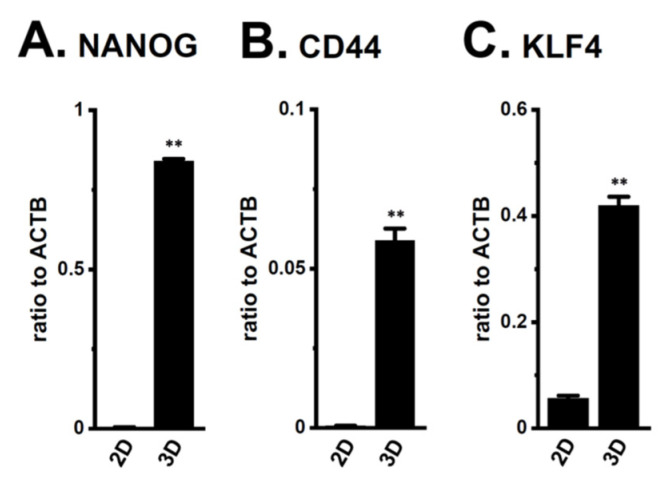
Expression of the transcripts of cancer stem markers, NANOG, CD44, and KLF4, in both 2D- and 3D-cultured LNCaP cells. (**A**–**C**): Real-time PCR examination of NANOG, CD44, and KLF4 in both groups (*n* = 4 for each). Expression levels were shown as a ratio to ACTB. **: *p* < 0.01 vs. ‘2D’.

**Figure 3 ijms-22-13553-f003:**
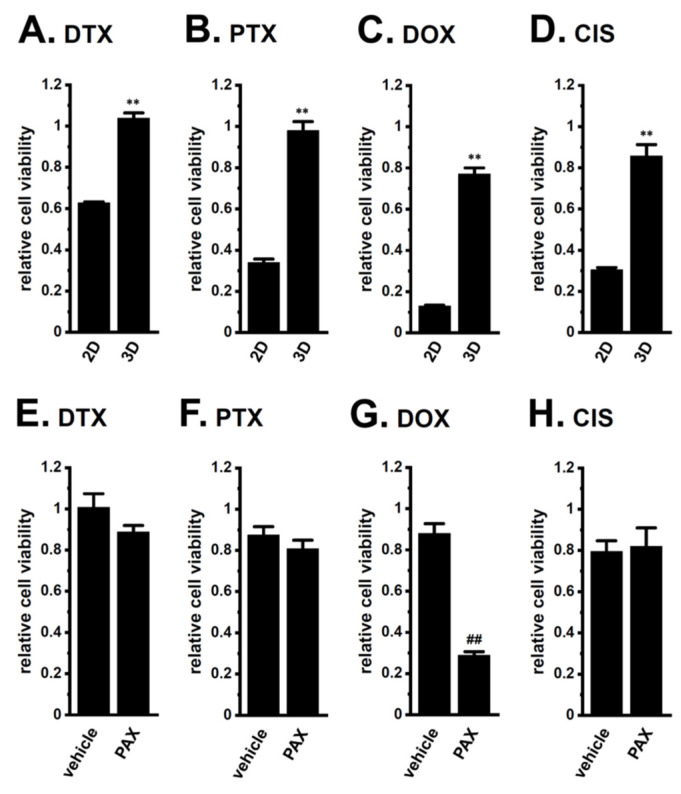
Effects of chemotherapy agents on the viability of 2D- and 3D-cultured LNCaP cells and the effects of a pretreatment with a K_Ca_1.1 inhibitor on chemoresistance acquired by 3D-cultured LNCaP cells. (**A**–**D**): Effects of the treatment with 100 nM DTX, 100 nM PTX, 1 μM DOX, and 10 μM CIS for 48 h on the viability of ‘2D’- and ‘3D’-cultured LNCaP cells using the WST-1 assay (*n* = 5 for each). Cell viability additing 0.1% dimethylsulfoxide, DMSO in DTX, PTX, and DOX and water in CIS instead of chemotherapy agents was expressed as 1.0. (**E**–**H**): Effects of the treatment with chemotherapy agents for 48 h on the viability of vehicle- and 10 μM PAX-pretreated (for 24 h), 3D-cultured LNCaP cells (*n* = 5 for each). **: *p* < 0.01 vs. ‘2D’; ^##^: *p* < 0.01 vs. vehicle control.

**Figure 4 ijms-22-13553-f004:**
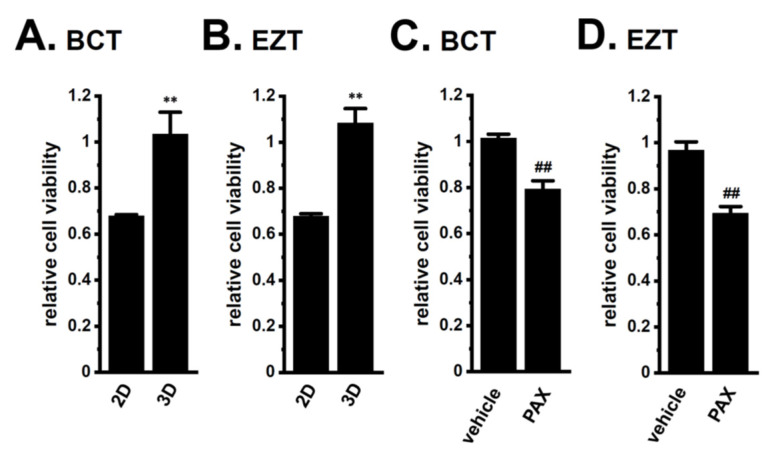
Effects of antiandrogens on the viability of 2D- and 3D-cultured LNCaP cells and the effects of a K_Ca_1.1 inhibitor on the antiandrogen resistance acquired by 3D-cultured LNCaP cells. (**A**,**B**): Effects of the treatment with 10 μM BCT and 10 μM EZT for 48 h on the viability of ‘2D’- and ‘3D’-cultured LNCaP cells using the WST-1 assay (*n* = 5 for each). Cell viability additing 0.1% DMSO instead of antiandrogens was expressed as 1.0. (**C**,**D**): Effects of the treatment with antiandrogens for 48 h on the viability of vehicle- and 10 μM PAX-pretreated (for 24 h), 3D-cultured LNCaP cells (*n* = 5 for each). **: *p* < 0.01 vs. ‘2D’; ^##^: *p* < 0.01 vs. vehicle control.

**Figure 5 ijms-22-13553-f005:**
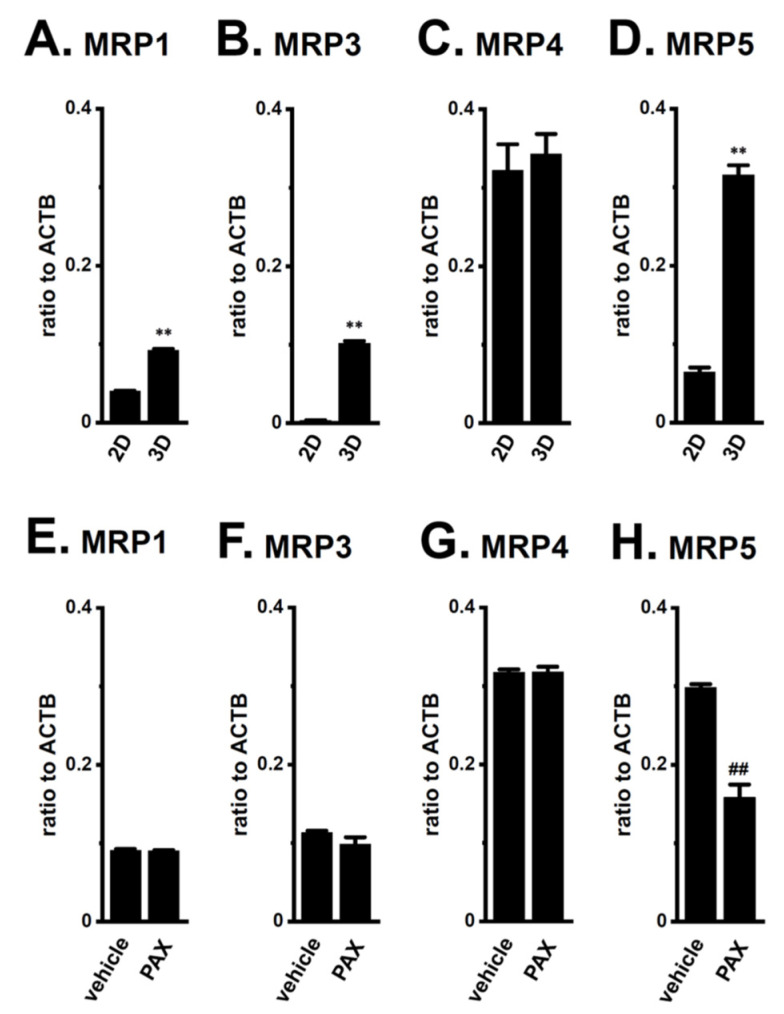
Expression of MRP transcripts in both 2D- and 3D-cultured LNCaP cells and the effects of K_Ca_1.1 blockade with PAX on their expression in 3D-cultured LNCaP cells. (A–D): Real-time PCR examination of MRP1 (**A**), MRP3 (**B**), MRP4 (**C**), and MRP5 (**D**) in ‘2D’ monolayers and ‘3D’ spheroids of LNCaP cells (*n* = 4 for each). (**E**–**H**): Real-time PCR examination of MRP transcripts in vehicle- and PAX-treated, 3D-cultured LNCaP cells (*n* = 4 for each). Expression levels were shown as a ratio to ACTB. **: *p* < 0.01 vs. ‘2D’; ^##^: *p* < 0.01 vs. vehicle control.

**Figure 6 ijms-22-13553-f006:**
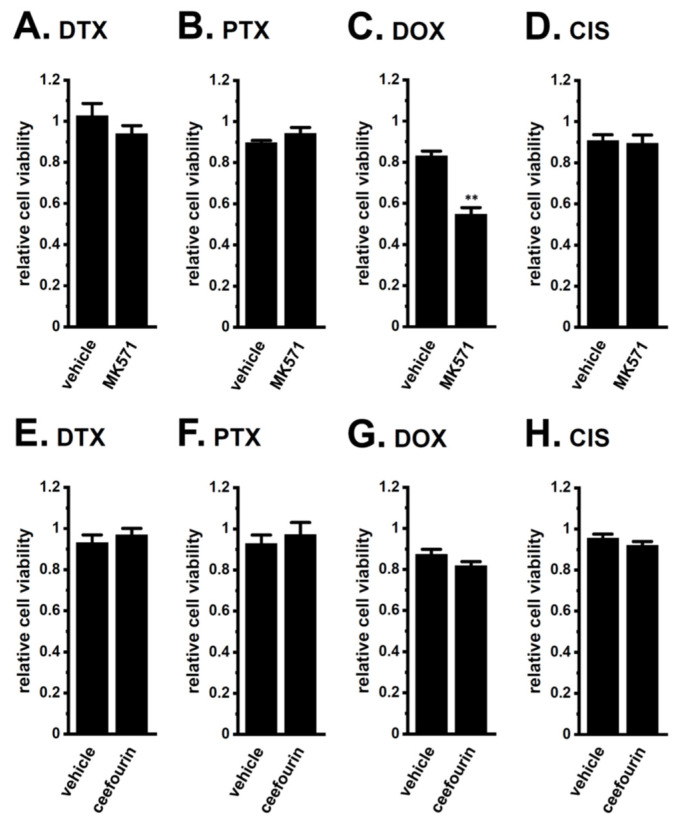
Effects of the MRP4/MRP5 inhibitor, MK571, and the selective MRP4 inhibitor, ceefourin 1, on chemoresistance acquired by 3D-cultured LNCaP cells. (**A**–**D**): Effects of the treatment with chemotherapy agents for 48 h on the viability of vehicle- and 10 μM MK571-cotreated, 3D-cultured LNCaP cells (*n* = 5 for each). (**E**–**H**): Effects of the treatment with chemotherapy agents for 48 h on the viability of vehicle- and 10 μM ceefourin 1-cotreated, 3D-cultured LNCaP cells (*n* = 5 for each). **: *p* < 0.01 vs. vehicle control.

**Figure 7 ijms-22-13553-f007:**
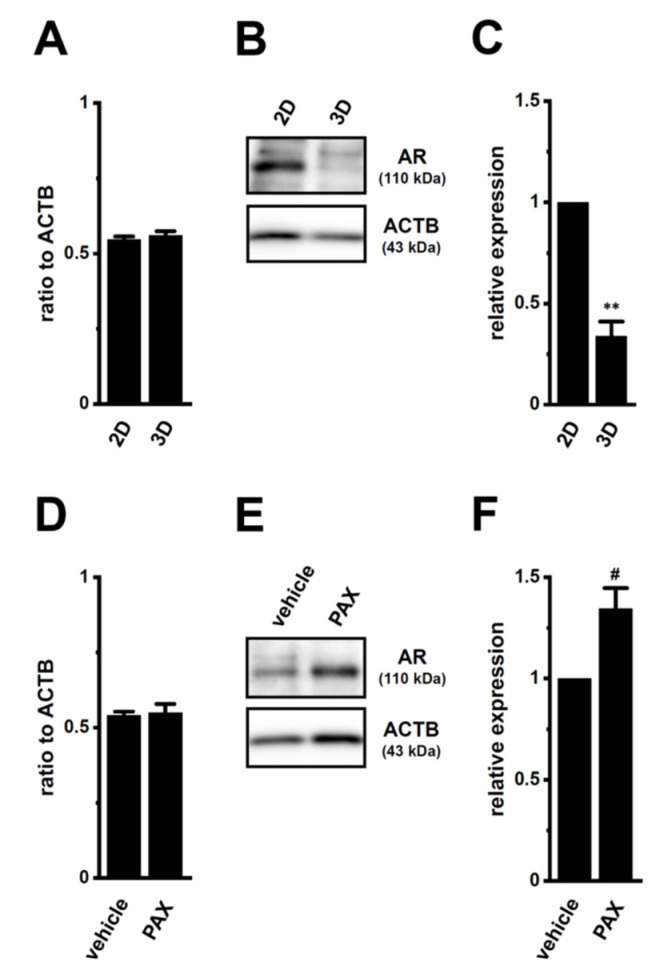
Decrease in AR protein expression by spheroid formation in LNCaP cells, and its reversal by the PAX treatment. (**A**,**D**): Real-time PCR examination of AR transcripts in 2D- and 3D-cultured LNCaP cells (**A**) and in vehicle- and PAX-treated (10 μM) (for 24 h), 3D-cultured LNCaP cells (**D**) (*n* = 4 for each). Expression levels are shown as a ratio to ACTB. (**B**,**E**): AR protein expression in protein lysates from lipid-raft-enriched fractions of 2D- and 3D-cultured LNCaP cells (**B**) and vehicle- and PAX-treated 3D-cultured LNCaP cells (**E**) (*n* = 4 for each). Blots were probed with anti-AR (approximately 110 kDa, upper panel) and anti-ACTB (43 kDa, lower panel) antibodies. (**C**,**F**): Summarized results were obtained as the optical density of AR and ACTB band signals, respectively (*n* = 4 for each). After compensation for the optical density of the AR protein band signal with that of the ACTB signal, the AR signal in ‘2D’ (**C**) or ‘vehicle’ (**F**) was expressed as 1.0 (*n* = 4 for each). **: *p* < 0.01 vs. ‘2D’; ^#^: *p* < 0.05 vs. vehicle control.

**Figure 8 ijms-22-13553-f008:**
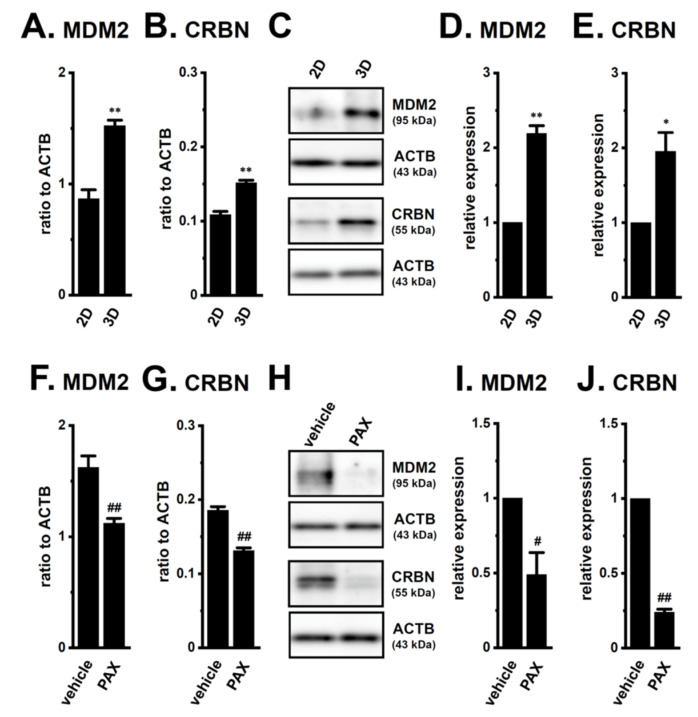
Comparison of expression levels of ubiquitin E3 ligases (MDM2 and CRBN) for AR between 2D- and 3D-cultured LNCaP cells, and effects of the PAX treatment in 3D-cultured LNCaP cells. (**A**,**B**): Real-time PCR examination of MDM2 (**A**) and CRBN (**B**) in ‘2D’- and ‘3D’-cultured LNCaP cells (*n* = 4 for each). Expression levels were shown as a ratio to ACTB. (**C**): Protein expression of MDM2 and CRBN in protein lysates from lipid-raft-enriched protein fractions of ‘2D’- and ‘3D’-cultured LNCaP cells. Blots were probed with anti-MDM2 (approximately 95 kDa), anti-CRBN (approximately 55 kDa), and anti-ACTB (43 kDa) antibodies. (**D**,**E**): Summarized results were obtained as the optical densities of MDM2 (**D**) and CRBN (**E**) band signals from ‘C’. After compensation for the optical densities of the protein band signals with that of the ACTB signal, the optical density in ‘2D’ was expressed as 1.0 (*n* = 4 for each). (**F**,**G**): Real-time PCR examination for MDM2 (**F**) and CRBN (**G**) in vehicle- and PAX-treated (10 μM) (for 24 h) LNCaP cells (*n* = 4 for each). (**H**): Protein expression of MDM2 and CRBN in protein lysates from lipid-raft-enriched protein fractions of vehicle- and PAX-treated LNCaP cells. (**I**,**J**): Summarized results were obtained as the optical densities of MDM2 (**I**) and CRBN (**J**) band signals from ‘H’ (*n* = 4 for each). *, **: *p* < 0.05. 0.01 vs. ‘2D’; ^#^, ^##^: *p* < 0.05. 0.01 vs. vehicle control.

**Figure 9 ijms-22-13553-f009:**
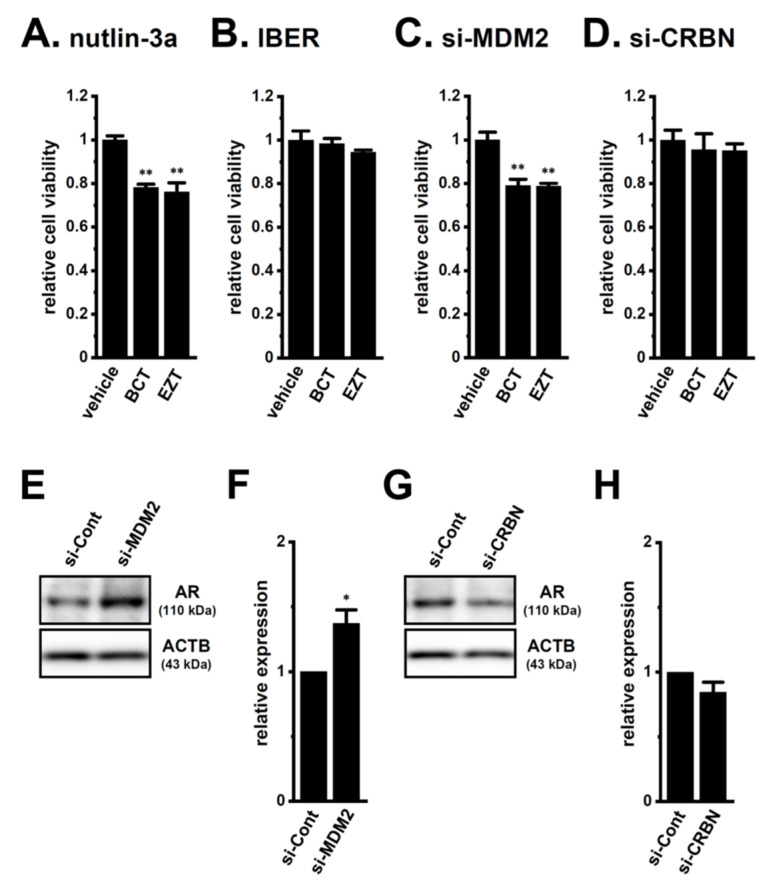
Effects of the pharmacological and siRNA-mediated inhibition of MDM2 and CRBN on the resistance to antiandrogens and AR protein expression in LNCaP spheroids. (**A**,**B**): Effects of the treatment with BCT and EZT (10 μM) for 48 h on the viability of 2 μM nutlin-3a- and 1 μM iberdomide (IBER)-pretreated (for 24 h) LNCaP spheroid cells (*n* = 5 for each). (**C**,**D**): Effects of the treatment with antiandrogens for 48 h on the viability of MDM2 siRNA (si-MDM2)- and CRBN siRNA (si-CRBN)-transfected, 3D-cultured LNCaP cells (*n* = 5 for each). (**E**,**G**): Effects of the siRNA-mediated MDM2 and CRBN inhibition on AR protein expression in protein lysates from the lipid-raft-enriched fraction of LNCaP spheroids. Blots were probed with anti-AR (approximately 110 kDa, upper panel) and anti-ACTB (43 kDa, lower panel) antibodies in the control siRNA (si-Cont)-, si-MDM2-, and si-CRBN-transfected groups. (**F**,**H**): Summarized results were obtained as the optical density of AR and ACTB band signals in si-Cont-, si-MDM2 (**F**)-, and si-CRBN (**H**)-transfected groups (*n* = 4 for each). After compensation for the optical density of the AR protein band signal with that of the ACTB signal, the AR signal in ‘si-Cont’ was expressed as 1.0 (*n* = 4 for each). *, **: *p* < 0.05, 0.01 vs. vehicle control and si-Cont.

**Figure 10 ijms-22-13553-f010:**
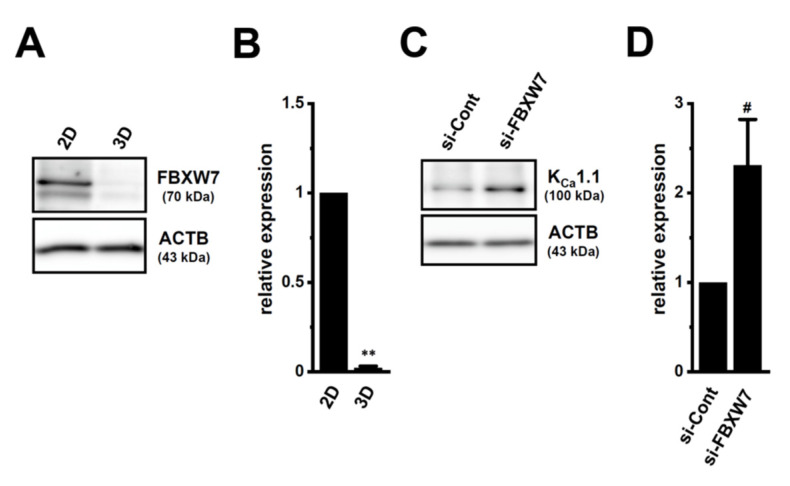
Regulation of K_Ca_1.1 protein degradation through the FBXW7 ubiquitin E3 ligase in LNCaP cells. (**A**,**B**) Protein expression of FBXW7 in whole cell protein lysates of ‘2D’- and ‘3D’-cultured LNCaP cells. Blots were probed with anti-FBXW7 (approximately 70 kDa) and anti-ACTB (43 kDa) antibodies (**A**). Summarized results (**B**) were obtained as the optical densities of FBXW7 and ACTB band signals (*n* = 4 for each). After compensation for the optical densities of the protein band signals with that of the ACTB signal, the optical density in ‘2D’ was expressed as 1.0 (*n* = 4 for each). (**C**,**D**): Protein expression of K_Ca_1.1 in control siRNA (si-Cont)- and FBXW7 siRNA (si-FBXW7)-transfected (for 72 h), 2D-cultured LNCaP cells. Blots were probed with anti-K_Ca_1.1 (approximately 100 kDa) and anti-ACTB (43 kDa) antibodies (**C**). Summarized results (**D**) were obtained as the optical densities of K_Ca_1.1 and ACTB band signals. The optical density in ‘si-Cont’ was expressed as 1.0 (*n* = 4 for each). **: *p* < 0.01 vs. ‘2D’; ^#^: *p* < 0.05 vs. ‘si-Cont’.

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
