# Peer review of "KCa1.1 K+ Channel Inhibition Overcomes Resistance to Antiandrogens and Doxorubicin in a Human Prostate Cancer LNCaP Spheroid Model"

_ijms, 2021, doi:10.3390/ijms222413553_

Round 1

Reviewer 1 Report

The manuscript is now improved. 

Reviewer 2 Report

Authors improved the manuscript according to my suggestions.

Reviewer 3 Report

The authors addressed my concerns 

This manuscript is a resubmission of an earlier submission. The following is a list of the peer review reports and author responses from that submission.

Round 1

Reviewer 1 Report

The authors present data to suggest that Spheroid culture of a single luminal-type prostate cancer cell line (LNCap) increases Kca1.1 activity when compared to conventional 2D cell culture. This result correlates with differential expression of genes associated with “stemness” when compared to 2D culture. The authors report that spheroid cultures of LNCaP show increased resistance to chemotherapeutic agents, which can be attenuated by inhibiting KCa1.1 function. One reason for the increased drug resistance of spheroid cultures to anti-androgens may be the reduced expression of AR, potentially through increased activation of E3 ligases.

The authors have clearly stated that their study involves only the investigation of LNCap spheroid cultures compared to (2D cultures). However, this severely reduces the impact of the work. At least the authors should repeat their findings using other luminal-type CaP cell lines for instance VCaP or DU-145. The paper would also benefit from some correlative assessment within existing human clinical databases to at least provide circumstantial evidence that the results presented may be applicable to human disease.

Specific points:

The authors should explain (or highlight more clearly) how cell numbers were normalized between 2D and 3D cell cultures. This will help readers understand how viability assays were controlled for cell number between the 2 methods of cell culture. This is especially important for the WST-1 assay as it measures metabolic activity therefore control of cell number is crucial.

Why did the authors not present their panel of stemness genes within figure 1, instead of adding some of them to sFig? All stemness genes could be efficiently presented on the same graph.

Fig 2A-D & 3A-D- Authors suggest that 3D cultures show increased chemoresistance compared to 2D cultures. For the benefit of readers, a graph should be added at least as SFig to demonstrate the viability changes after treatment within each 2D or 3D culture. I.e. include the base line viability (vehicle) and relative change after treatment for 2D and 3D cultures.  Otherwise it is not immediately clear what drives the changes.

6B. Since the Blot shows almost no actin in the 3D culture sample it is impossible to conclude that AR expression is decreased. It is clear that the sample loading is much lower in the 3D culture. Given that 6B is a representative image for scanning densitometry of multiple samples depicted in C, then it suggests that experimental replicates had the same issue. Scanning densitometry is a useful way to represent data only if the effect is clear. A more suitable blot is needed, preferably depicting multiple replicates in order to convince this reviewer.

General points: for western blots with accompanying scanning densitometry, why did the authors not run all samples on the same gel? If they did (as is necessary for densitometry analysis) then they should present all samples as WB images.  

Reviewer 2 Report

The submitted manuscript assessed the potential of the KCa1.1 K+ Channel as a therapeutic target to overcome enzalutamide resistance in metastasized prostate cancer (mPCa). This study addresses a current problem for patients with mPCa. The manuscript is written clearly and fairly. The authors appropriately cite previous literature and consider their results in light of the current knowledge in the field. The conclusions are clearly stated and supported by the data. While most of the paper is very well written, there are numerous issues and inaccuracies. These should be addressed before acceptance:

  1. All experiments have only been performed in the hormone-sensitive cell line LNCaP. At least the main results should be repeated in another cell line. Preferable in a castration-resistant cell model.
  2. The authors claim that inhibition of the KCa1.1 K+ Channel overcomes enzalutamide resistance. Therefore, the inhibition should be tested in an enzalutamide-resistant cell model.
  3. In Fig.1, only NANOG has been shown, whereas CD44 and KL4F have been moved to the supplements. Therefore, there is no reason why NANOG is more important than the other markers for cancer stemness, all be shown in the main figure.
  4. In line 178, the authors report that only 4 out of 10 ABC transporter are highly expressed. However, the results of the other ABC transporters. Moreover, as shown in Figure 1D+E, protein expression can change without a change in mRNA. Therefore, expression experiments should be performed on mRNA and protein levels.
  5. In general, data not shown should not been used and all mentioned experiments should be shown at least in the supplementary data.
  6. Western Blot and qPCR for siRNA efficiency have to be shown.
  7. Please add the Used Antibody dilutions into the methods section
  8. All uncropped western blots should be shown in the supplementary data
  9. Minor comment: Current cancer statistics and guidelines should be used in the introduction. In the treatment options for CRPC, darolutamide, apalutamide, PARP inhibitors, and Lu-PSMA should be mentioned.
  10. Minor comment: As the authors mention, AR-degradation current studies such as PMID: 26257066, PMID: 21216933n, or PMID: 21216933 should be discussed.

Reviewer 3 Report

In the present manuscript, Dr. Ohya and colleagues generate spheroids derived by LNCaP cells. After specific steps, these spheroids acquire antiandrogen resistance. The goal of their manuscript is to show that the inhibition of KCa1.1, a Ca2+-activated K+ can overcome the resistance to antiandrogens. 

Here, my concerns follow:

In the introduction section, the authors write "They are a valuable tool for studying the tumor microenvironment (TME) in solid tumors". They should discuss the role of the tumor microenvironment in Prostate cancer (PC) with particular attention to the role of androgen receptor (AR; doi: 10.1101/gad.315739.118; doi: 10.1038/s41419-021-03402-7).

Results:

-the experiments are based only on LNCaP cells. Androgen deprivation therapy (ADT) is a first-line treatment in the management of advanced prostate cancer (PC). The authors write that "Recent studies demonstrated that ion channels are important contributors to drug resistance being overcome in solid cancers". Nevertheless, they only perform experiments in LNCaP cells that are Androgen sensitive. Also if they promote acquired resistance in LNCaP cells in order to accept the author's conclusions, it would be useful to check the effects in C4-2 cells in which there is a high basal expression and activity of the androgen receptor, and in addition in DU-145 and/or PC3 cells that are AR-negative and insensitive to androgens. 

-In addition, what are the effects of their study on non-tumor prostate-derived cells (such as RWPE1) ?

The authors provide the formation of spheroids in ultra-low attachment plates, but what happens if they use a component of the extracellular matrix? It is known that matrix is important in 3D models and that, when used, the spheroids may warn otherwise the drugs used.  It could be intriguing.

At last, the authors should modify some phrases since they are gray and not black. In addition, they should re-write their manuscript in a more fluent way.